

# Association between lung function impairment with urinary heavy metals in a community in Klang Valley, Malaysia

Ammar Amsyar Abdul Haddi[1,2], Mohd Hasni Ja'afar[1] and Halim Ismail[1]

[1] Department of Community Health, Faculty of Medicine, Universiti Kebangsaan Malaysia, Cheras, Kuala Lumpur, Malaysia

[2] Ministry of Health, Putrajaya, Malaysia

## ABSTRACT

Lung function status can be directly or indirectly affected by exposure to pollutants in the environment. Urinary heavy metals may be an indirect indicator of lung function impairment that leads to various diseases such as chronic obstructive pulmonary disease (COPD). This study aimed to explore the prevalence of lung function impairment as well as its association with urinary heavy metal levels and other influencing factors among the community in Klang Valley, Malaysia. Urinary sampling was done during various community events in the housing areas of Klang Valley between March and October 2019. Only respondents who consented would undergo a lung function test. Urine samples were obtained and sent for Inductively Coupled Plasma Mass Spectrometry (ICP-MS) analysis for heavy metal cadmium (Cd) and lead (Pb) concentration. Of the 200 recruited respondents, 52% were male and their ages ranged from 18 years old to 74 years old with a mean age of $38.4 \pm 14.05$ years. Urinary samples show high urinary Cd level in 12% of the respondents ($n = 24$) whereas none recorded a high urinary Pb level. There was a positive correlation between the levels of urinary Cd and urinary Pb ($r = 0.303$; $p = 0.001$). Furthermore, a negative correlation was detected between urinary Cd level and forced vital capacity (FVC) ($r = -0.202$, $p = 0.004$), force expiratory volume at the first second (FEV1) ($r = -0.225$, $p = 0.001$), and also force expiratory flow between 25–75% of FVC (FEF 25–75%) ($r = -0.187$, $p = 0.008$). However, urinary Pb did not show any correlation with lung function parameters. Multiple linear regression analysis showed that urinary Cd had a significant negative effect on FVC ($p = 0.025$) and FEV1 ($p = 0.004$) based on the predicted value. Additionally, other factors such as education level ($p = 0.013$) also influenced lung function. However, no interaction was detected between heavy metals or other factors. In short, there was a significant negative linear relationship between urinary Cd and lung function, whereas urinary Pb was not associated with lung function. Beside acting as a biomarker for cadmium exposure level, urinary Cd may also be applied as indirect biomarker for asymptomatic chronic lung function deterioration among the healthy population.

Corresponding authors
Ammar Amsyar Abdul Haddi, ammar353548@yahoo.com
Mohd Hasni Ja'afar, drmhasni1965@gmail.com, drmhasni@ukm.edu.my

## INTRODUCTION

Lung diseases are characterized by poor lung function status and the severity of lung impairment can be determined using certain parameters. Lung function is used as a screening tool for underlying lung impairment. It also acts as a diagnostic tool for lung problems like chronic obstructive pulmonary disease (COPD) (GOLD, 2010). COPD is a debilitating lung disease that has affected at least 12% of the world population. It causes a tremendous impact on the patient, family, and nation as a result of direct and indirect costs due to complications of COPD and the subsequent productivity losses (*López-Campos, Tan & Soriano, 2016*). In Malaysia, the prevalence of COPD was estimated to be 6.5% (*Loh et al., 2016*) with a productivity loss of at least 2,200 USD per patient per annum due to the direct and indirect costs of COPD (*Ur Rehman et al., 2021*).

Lung impairment such as COPD has been linked with environmental factors such as lifestyle and exposure to air pollutants (*Bai et al., 2017*). One of the major pollutants that adversely impair lung function is heavy metals, particularly those present in concentrated form in tobacco products such as cadmium (Cd) and lead (Pb) (*Engida & Chandravanshi, 2017*). Indirectly, cadmium and lead may damage lung interstitial tissues and cells by triggering an inflammatory response, thus leading to COPD (*Cabral et al., 2015*; *Sundblad et al., 2016*). On top of tobacco cigarettes being the main source of exposure, heavy metals are also commonly detected in ambient air and food (*Jeevanaraj et al., 2020*), bottled drinking water (*Mohd Hasni et al., 2017*), and road dust (*Wahab et al., 2020*). The link between heavy metals such as Cd and Pb with respiratory problems has also been reported among children with a higher risk of reported respiratory problems, as evidenced by the elevated heavy metals levels in their fingernails (*Esphylin et al., 2018*).

With an aging population and an increase in the smoking prevalence, there is a possibility that the prevalence of poor lung function and COPD will steadily rise. Despite cigarette smoking being known as the main source of heavy metals, there is a need to explore other confounding factors that may associated with heavy metals and lung function. Therefore, this study aimed to determine the lung function level and urinary heavy metals level among the Klang Valley residents and to look for possible associations between urinary levels of lead and cadmium with lung function impairment.

## METHODS & MATERIALS

This cross-sectional study was conducted among the general population in the Klang Valley, an urbanized area in Malaysia. Inclusion criteria were adult Malaysian citizens 18 years old and above who lived and worked in the Klang Valley for at least 3 years as well as agreed to participate with written informed consent. Those with a barrier and difficulties in performing lung function tests or producing urine samples were excluded, in addition to participant presented with respiratory tract infection symptoms or had active exacerbation of Asthma or COPD were excluded as well.

Samples were recruited *via* multistage sampling, *i.e.,* purposive sampling followed by systematic random sampling. In the first stage, purposive sampling of community events that were held between March and November 2019 in the Klang Valley public places was

performed. A total of ten events were selected. The registration list of the community event was then used as a sampling frame for the second phase of systematic random sampling to recruit the study samples based on the participants' registration number on the event's registration list.

The sample size was calculated using the Power sample size calculator developed by *Dupont & Plummer Jr (1998)* based on power of 80% and beta error of 0.05. A total of 196 samples was required. Data collection started with a self-administered questionnaire on basic sociodemographic, socioeconomic, and lifestyle data, followed by anthropometric measurements (weight and height). This was followed by urine specimen collection and lastly, lung function test under the guidance of the researcher.

## Tools

Anthropometric measurement, *i.e.,* height was measured using the SECA 217 Stadiometer from Japan whereas weight was measured by using the SECA 813 Digital High-Capacity Floor Scale, Japan. Both instruments were calibrated regularly prior to the event.

The urine specimen was collected using a clean catch, mid-stream technique with proper explanation was given to the participant before the procedure. The urine sample was stored in a disposable urine container and being stored in the fridge below freezing temperature before analysis. Urinary heavy metals were analyzed using the Inductive Coupled Plasma Mass Spectrometry (ICP-MS) machine model Perkin Elmer ELAN 9000 (Waltham, MA, USA) and analyzed using the Graphite Furnace Atomic Absorption Spectrometry (GFAAS) method. The lowest detection limit of the equipment was 0.1 µg/L.

The tool used for lung function measurement was the spirometry model CHESTGRAPH HI-105 by CHEST Inc. (Tokyo, Japan). The faculty of Medicine, University Kebangsaan Malaysia (UKM) owned and regularly calibrated the machine. The operator conducting the spirometric lung function was trained personnel. All participants were briefed with a visual demonstration on the correct way to perform the spirometry test. The lung function test was conducted and interpreted based on the guidelines of the American Thoracic Society (*Graham et al., 2019*). Three satisfactory attempts were recorded from each respondent to obtain the best result.

## Statistical analysis

The data were analyzed using Statistical Package for Social Science (SPSS) Version 23. Primary outcome variables and lung function parameters were analyzed in continuous form. For categorical data, normally distributed data were presented as mean and standard deviation (SD) whereas non-normally distributed data were presented as median and interquartile range (IQR). All continuous data were checked for normality. Non-normally distributed data including urinary Cd and Pb levels were transformed into normal distribution by using log transformation for urinary Cd before inferential analysis, whereas square root transformation method was used for urinary Pb instead due to log transformation method was unsuccessful. Linear regression and Pearson's correlation test were used to analyze the correlation between continuous variables on lung function whereas independent *t*-test and one-way ANOVA test were used for the analysis of

categorical variables against the lung function. Multiple linear regression was performed to determine the predictor of lung function. The rational for exploring correlation between lung function and other covariables was aimed to identify confounding factors in the multivariable regression analysis.

## Definition

In this study, a high level of urinary Cd was defined by having a urinary Cd concentration of more than 2 µg/L (*Ke et al., 2015*) whereas high urinary Pb level was defined by having a urinary Pb concentration of above 50 µg/L according to the United States Center of Disease Prevention guideline (*Abadin et al., 2019*).

Lung function outcomes were determined with seven parameters, namely FEV1 (Forced Expiratory Volume in the first second): the volume of air that the patient can exhale in the first second of forced expiration; FVC (Forced Vital Capacity): the total volume of air that the patient can forcibly exhale in one breath; FEV1/ FVC ratio (the ratio of FEV1 to FVC expressed as a fraction or ratio); FEV1% (percentage of FEV1 achieved when compared to predicted value based on given age, weight, height, and race; FVC% (percentage of FVC achieved when compared to predicted value based on given age, weight and height; FEF 25–75% (forced expiratory flow averaged over the middle portion of FVC, specifically between 25% and 75% of the FVC), also known as maximal mid-expiratory flow; lastly PEF (peak expiratory flow rate) that measures a person's maximum speed of expiration in the form of volume of air against time. FVC, FEV1, and FEF 25–75% were expressed in Liter; FEV1% and FVC% were expressed in percentage; FEV1/FVC expressed in a ratio; and PEF in L/ second.

The presence of "lung function impairment" was defined as having any of these criteria: (a) FEV1% predicted score of less than 70%, (b) FVC% predicted score of less than 70%, or (c) FEV1/ FVC ratio score of less than 0.7 (*GOLD, 2010*; *Mehrparvar et al., 2014*).

In addition, the variable smoking was defined by the "current smoker" status based on the GATS questionnaire (*WHO, 2011*). Air quality was categorized as either PM10 and PM2.5 based on the data obtained from the nearest air quality monitoring station of the Department of Environment. The air quality data referred to the mean value of a 3-month concentration of PM10 and PM2.5 before the sampling date in µg/m$^3$ (*Hashemzadeh et al., 2019*).

This study received ethical approval from the University Kebangsaan Malaysia (UKM) Research Ethical Committee (Ethical Approval reference: UKM PPI/111/8/JEP-2018699 FF-2019-043, 10 January 2019).

## RESULTS

A total of 200 people participated in this study (Table 1). They were more than half males (52.0%), Malays (87.0%), employed (66.0%), with tertiary level of education (59.5%). The mean age of the participants was 38.4 years old. The mean year lived in the Klang Valley was 23.4 years. The participants had a median household income of RM 3000. About 44.0% of the respondents lived in low-cost flats. One-quarter (26.0%) of respondents were smokers with a mean duration of smoking of 20.7 years. The mean cigarette smoked per

**Table 1  Baseline characteristics of participants.**

| Variables | n (%) | Mean ± SD | Median (IQR) |
|---|---|---|---|
| **Age (year)** | | 38.4 ± 14.05) | |
| **Gender** | | | |
| Male | 104 (52.0) | | |
| Female | 96 (48.0) | | |
| **Ethnicity** | | | |
| Malay | 174 (87.0) | | |
| Non-Malay | 26 (13.0) | | |
| **Employment status** | | | |
| Employed | 132 (66.0) | | |
| Unemployed | 68 (34.0) | | |
| **Household income (RM)** | | | 3000 (1750–3500) |
| **Highest education level** | | | |
| Primary | 12 (6.0) | | |
| Secondary | 69 (34.5) | | |
| Tertiary | 119 (59.5) | | |
| **Years living in Klang Valley** | | 23.5 (±13.16) | |
| **Hobby/job related to oil painting** | | | |
| Yes | 14 (7.0) | | |
| No | 186 (93.0) | | |
| **Hobby/job related to gardening** | | | |
| Yes | 58 (29.0) | | |
| No | 142 (71.0) | | |
| **Smoking** | | | |
| Yes | 50 (25.0) | | |
| No | 150 (75.0) | | |
| **Duration of smoking (years)** | | 20.4 (±13.09) | |
| **Cigarette smoked per day** | | 10.0 (±5.94) | |
| **Body mass index** | | 26.8 (±5.95) | |
| **3-month average PM10 ($\mu$g/m$^3$)** | | 32.6 (±5.16) | |
| **3-month average PM2.5 ($\mu$g/m$^3$)** | | 23.7 (±2.69) | |

day among them was 10 cigarettes. About 29.0% of the participants were involved in a job/hobby related to gardening whereas 7.0% had a hobby/job related to painting.

The anthropometric parameters such as height, weight, and body mass index (BMI) were normally distributed with means of 1.6 m, 71.2 kg, and 26.8 respectively A three-month average of PM10 and PM2.5 concentrations were 34.8 $\mu$g/m$^3$ and 23.7 $\mu$g/m$^3$ respectively. The data of urinary heavy metals were not normally distributed. The median urinary Cd and Pb concentrations were 1.0 $\mu$g/L and 5.9 $\mu$g/L respectively. About 12.0% of participants had urinary Cd above the threshold level whereas none of them showed urinary Pb above the threshold level (Table 2).

For lung function parameters, the mean FVC score among participants was 3.0 L whereas the mean FEV1 score among participants was 2.5 L. The mean FVC% achieved from the

**Table 2** Level of urinary heavy metals and lung function among participants ($n = 200$).

| Parameter | Median (IQR) | Mean ± SD | n (%) of samples with above-threshold level | |
|---|---|---|---|---|
| | | | Cd | Pb |
| **Urinary Cd (µg/L)** | 0.90 (0.06–0.14) | 1.20 ± 1.04 | 24 (12.0) | 0 (0.0) |
| **Urinary Pb (µg/L)** | 5.90 (4.64–7.36) | 6.90 ± 5.14 | | |
| FVC (L) | | 3.00 ± 0.82 | | |
| FEV1 (L) | | 2.50 ± 0.72 | | |
| FEV1/ FVC | 0.80 (0.79–0.87) | | | |
| % FVC (%) | | 78.40 ± 11.84 | | |
| % FEV1 (%) | | 79.80 ± 13.04 | | |
| FEF 25–75% (L/hour) | | 2.80 ± 1.12 | | |
| PEF (L/ hour) | | 6.50 ± 2.41 | | |

predicted value was 78.4% whereas the mean FEV1% achieved from the predicted value was 79.5%. In addition, the median FEV1/ FVC ratio among the participants was 0.8 whereas the mean FEF 25–75% and PEF score among participants were 2.8 L and 6.5 L/minute respectively. In terms of lung function impairment, it was noted that 18.7%, 17.2%, and 3.0% of participants showed impaired FVC, FEV1, and FEV1/ FVC ratio respectively.

After identifying and removing the outliers, 198 participants were included in the correlation analysis between urinary levels of Cd, Pb, and lung function (Table 3). All variables were analyzed using Pearson's correlation except for FEV1/FVC (Spearman's correlation) as the data was not normally distributed. Log urinary Cd was significantly and negatively correlated with FVC, FEV1, % FVC, % FEV1, and FEF 25–75% ($p$-value < 0.05) but the strength of correlations was considered weak. On the other hand, FEV1/ FVC and PEF were not correlated with log urinary Cd. Meanwhile, the square root of urinary Pb was only weakly correlated with PEF ($p < 0.05$) and it did not correlate with other lung function parameters. Lastly, the Log urinary Cd and square root urinary Pb also showed a fairly significant positive correlation.

Additionally, the correlation analysis between lung function and other covariates showed that age was associated with almost all the lung function parameters (except for %FEV1 & PEF) in a negative linear relationship ($p$-value < 0.05) (Table 4). On the other hand, in a positive linear relationship, log household income was associated with almost all lung function parameters, except for FEV1/ FVC ($p$-value < 0.05). In addition, the duration of having lived in the Klang Valley (in years) was also significantly correlated with FVC, FEV1, FEF 25–75%, PEF negatively, with a strong correlation noted for FEV1 (−.362).

Next, the correlation analysis between lung function and air quality shows that PM10 and PM2.5 were weakly and negatively correlated only with PEF ($p = 0.012$, $p = 0.037$). FVC was also negatively correlated with PM2.5 ($p = 0.022$). A negative correlation was also observed with other parameters but it was not significant. Although the strength of correlation was weak, BMI was found to be negatively correlated with FEV1/ FVC ($p = 0.02$) but positively correlated with PEF ($p = 0.012$).

**Table 3** Correlation analysis between urinary heavy metals and lung function ($n = 198$) using Pearson's correlation.

| Variable | $p$-value | R value | $r^2$ |
|---|---|---|---|
| **Log Urinary Cd** | | | |
| FVC | 0.004 | −0.202 | 0.041 |
| FEV1 | 0.001 | −0.225 | 0.051 |
| FEV$_1$/FVC[*] | 0.061[*] | −0.133 | 0.018 |
| % FVC | 0.001 | −0.227 | 0.052 |
| % FEV1 | 0.007 | −0.141 | 0.020 |
| FEF 25–75% | 0.008 | −0.187 | 0.035 |
| PEF | 0.186 | −0.094 | 0.009 |
| **Square root Urinary Pb** | | | |
| FVC | 0.368 | 0.064 | 0.004 |
| FEV$_1$ | 0.277 | 0.078 | 0.006 |
| FEV$_1$/FVC[*] | 0.802[*] | 0.016 | <0.001 |
| % FVC | 0.502 | −0.048 | 0.002 |
| % FEV1 | 0.958 | 0.004 | <0.001 |
| FEF 25–75% | 0.400 | 0.060 | 0.004 |
| PEF | 0.030 | 0.154 | 0.024 |
| **Log Urinary Cd** | | | |
| Square root Urinary Pb | 0.001 | 0.303 | 0.092 |

**Notes.**

r value indicates strength of correlation, negative (-) value indicates negative correlation.

*FEV1/FVC was analyzed using Spearman's correlation due to non-normal distribution.

Correlation analysis between smoking and lung function also shows negative linear relationship between smoking duration in years and most of the lung function parameters ($p$-value < 0.05) whereby the strongest correlation was seen in FEF 25–75% (r = −.618). In contrast, the number of cigarettes smoked per day did not significantly correlate with any lung function parameters (Table 4).

Bivariate analysis revealed that gender, employment status, ethnicity, and education level were associated with certain lung function parameters (Tables 5 and 6). In addition, education level was associated with all the lung function parameters except PEF.

In addition, gender was associated with all the lung function parameters. In general, males fared better. Ethnicity-wise Malays recorded better % FVC, %FEV1, and FEF 25–75% scores compared to non-Malays. Being unemployed also gave a lower score of FVC, FEV1, FEF 25–75%, and PEF.

For the correlation analysis between urinary heavy metals and other socio-environmental factors, urinary Cd was significantly correlated with age, log household income, education level, BMI, 3-month average PM10 and PM2.5, as well as smoking duration. Similarly, urinary Pb was significantly correlated with 3-month average PM10 and PM2.5, and the number of cigarettes smoked per day (Table 7).

A significant difference was detected between the mean of log urinary Cd and the variable of hobby/ job related to gardening, and education level. As for the square root urinary Pb, it showed a significant difference with ethnicity and employment status (Table 8).

**Table 4** Correlation analysis between lung function and sociodemographic data, air quality, body mass index, and smoking behavior using Pearson's correlation.

| Variable | $p$-value | r value | $r^2$ |
|---|---|---|---|
| **Age ($n = 198$)** | | | |
| FVC | <0.001 | −.385 | 0.148 |
| FEV1 | <0.001 | −.498 | 0.248 |
| $FEV_1$/FVC | <0.001* | −.432 | 0.187 |
| %FVC | 0.002 | −.223 | 0.050 |
| %FEV1 | 0.099 | −.192 | 0.037 |
| FEF25–75% | <0.001 | −.481 | 0.231 |
| PEF | 0.092 | −.137 | 0.019 |
| **Log household income ($n = 150$)** | | | |
| FVC | <0.001 | .287 | 0.082 |
| $FEV_1$ | 0.001 | .269 | 0.072 |
| $FEV_1$/FVC | 0.422 | −.057 | 0.003 |
| %FVC | 0.023 | .185 | 0.034 |
| %FEV1 | 0.043 | .166 | 0.028 |
| FEF25–75% | 0.021 | .188 | 0.035 |
| PEF | <0.001 | .305 | 0.093 |
| **Years of living in Klang Valley** | ($n = 192$) | | |
| FVC | <0.001 | −.301 | 0.091 |
| FEV1 | <0.001 | −.362 | 0.131 |
| $FEV_1$/FVC | 0.011 | −.231 | 0.053 |
| %FVC | 0.509 | −.048 | 0.002 |
| %FEV1 | 0.566 | −.042 | 0.002 |
| FEF25–75% | <0.001 | −.344 | 0.118 |
| PEF | 0.039 | −.149 | 0.022 |
| **3-month average PM10 ($N = 198$)** | | | |
| FVC | 0.067 | −.130 | 0.017 |
| FEV1 | 0.142 | −.105 | 0.011 |
| $FEV_1$/FVC | 0.027 | −.158 | 0.025 |
| % FVC | 0.839 | .025 | 0.001 |
| % FEV1 | 0.559 | .038 | 0.001 |
| FEF 25–75% | 0.956 | .011 | <0.001 |
| PEF | 0.012 | −.177 | 0.031 |
| **3-month average PM2.5 ($N = 198$)** | | | |
| FVC | 0.022 | −.163 | 0.027 |
| $FEV_1$ | 0.298 | −.131 | 0.017 |
| $FEV_1$/FVC | 0.207 | .090 | 0.008 |
| % FVC | 0.673 | .030 | 0.001 |
| % FEV1 | 0.548 | .043 | 0.002 |
| FEF 25–75% | 0.978 | .002 | <0.001 |
| PEF | 0.037 | −.148 | 0.022 |

**Table 4** (*continued*)

| Variable | *p*-value | r value | r$^2$ |
|---|---|---|---|
| **Body mass index** ($N = 198$) | | | |
| FVC | 0.237 | .085 | 0.007 |
| FEV$_1$ | 0.652 | .032 | 0.001 |
| FEV$_1$/FVC | 0.020 | −.165 | 0.027 |
| % FVC | 0.257 | .081 | 0.007 |
| % FEV1 | 0.443 | .055 | 0.003 |
| FEF 25–75% | 1.000 | <.001 | <0.001 |
| PEF | 0.012 | .179 | 0.032 |
| **Smoking Duration (years)** ($N = 50$) | | | |
| FVC | 0.001 | −.447 | 0.200 |
| FEV1 | <0.001 | −.584 | 0.341 |
| FEV$_1$/FVC* | <0.001* | −.530 | 0.281 |
| %FVC | 0.097 | −.237 | 0.067 |
| %FEV1 | 0.025 | −.317 | 0.101 |
| FEF25–75% | <0.001 | −.618 | 0.425 |
| PEF | 0.068 | −.260 | 0.068 |
| **Number of cigarettes smoked/day** ($N = 50$) | | | |
| FVC | 0.873 | −.023 | 0.005 |
| FEV$_1$ | 0.329 | −.141 | 0.020 |
| FEV$_1$/FVC* | 0.208* | −.181 | 0.033 |
| % FVC | 0.841 | −.029 | 0.001 |
| % FEV1 | 0.526 | −.092 | 0.009 |
| FEF 25–75% | 0.117 | −.225 | 0.051 |
| PEF | 0.519 | −.093 | 0.009 |

**Notes.**
  *FEV1/FVC was analyzed using Spearman's correlation due to non-normal distribution.

## Multivariate analysis

After controlling for confounders such as smoking and gender, urinary Cd remained a significant predictor of lower FEV1%. In addition, tertiary education level and Malay were also significant predictors of higher FEV1% (Table 9). This model detected no interaction between urinary Cd and ethnicity or education level.

   In the FVC% final model using multiple linear regression analysis, urinary Cd was significantly associated with FVC% in a negative linear relationship. Age and ethnicity (non-Malay) were also significant predictors of lower %FVC. No interaction was detected in this model.

   In the final regression model for urinary Cd and Pb, the 3-month average PM2.5 concentration was identified as a predictor of both urinary Cd and Pb. In addition, age was a significant predictor of urinary Cd (Table 10).

## DISCUSSION

In this study, the mean FVC and FEV$_1$ were similar to previous studies (*Abdullah et al., 2018*; *Bandyopadhyay, 2011*). About one in five (18.7%) of the participants showed impaired FVC% value whereas 17.2% of samples recorded impaired FEV1% value. The

**Table 5** Mean differences in lung function by education level using one-way ANOVA test.

| Variable | | n | Mean (SD) | df | F | p-value |
|---|---|---|---|---|---|---|
| **Education level** | | | | | | |
| **FVC** | **Primary** | 12 | 2.5 (0.817) | | | |
| | Secondary | 69 | 2.8 (0.762) | | | |
| | Tertiary | 117 | 3.2 (0.809) | | | |
| | | | | 2,195 | 7.733 | 0.001 |
| **FEV1** | **Primary** | 12 | 2.0 (0.684) | | | |
| | Secondary | 69 | 2.2 (0.653) | | | |
| | Tertiary | 117 | 2.7 (0.692) | | | |
| | | | | 2,195 | 12.60 | <0.001 |
| **FEV1/FVC**[**] | **Primary** | 12 | 0.8 (0.736) | | | |
| | Secondary | 69 | 0.9 (0.986) | | | |
| | Tertiary | 117 | 0.9 (0.616) | | | |
| | | | | 2 | 12.73 | 0.002 |
| **%FVC** | **Primary** | 12 | 74.4 (11.02) | | | |
| | Secondary | 69 | 75.4 (11.72) | | | |
| | Tertiary | 117 | 80.6 (11.58) | | | |
| | | | | 2,195 | 5.132 | 0.007 |
| **%FEV1** | **Primary** | 12 | 74.3 (11.11) | | | |
| | Secondary | 69 | 76.5 (13.96) | | | |
| | Tertiary | 117 | 82.2 (12.13) | | | |
| | | | | 2,195 | 5.609 | 0.004 |
| **FEF 25–75%** | **Primary** | 12 | 2.2 (1.04) | | | |
| | Secondary | 69 | 2.4 (1.05) | | | |
| | Tertiary | 117 | 3.1 (1.08) | | | |
| | | | | 2,195 | 13.10 | <0.000 |
| **PEF** | **Primary** | 12 | 5.2 (2.69) | | | |
| | Secondary | 69 | 6.4 (2.37) | | | |
| | Tertiary | 117 | 6.7 (2.37) | | | |
| | | | | 2,195 | 2.205 | 0.113 |

**Notes.**
df = degree of freedom; (within group, between group).
*FEV1/FVC was analyzed using Kruskal-Wallis test due to non-normal distribution.

reported prevalence of lung impairment was similar to the prevalence in the United States (US) (*Schwartz et al., 2020*). However, it showed an increasing trend locally as compared to the previously published study with a prevalence of 15.7% (*Sui et al., 2015*). Moreover, 3.0% of the participant in this study recorded an $FEV_1/FVC$ ratio of below 0.7, much lower than the national prevalence of COPD (*Loh et al., 2016*). However, a survey done in China showed a higher prevalence of borderline lung function impairment at 43.0% (*Xiao et al., 2020*). Generally speaking, the prevalence of lung function impairment was far higher than the prevalence of COPD, thus indicating that certain lung function impairment, especially restrictive impairment was often underestimated and underappreciated. Therefore, clinicians have a widespread call for this issue to be taken seriously (*Godfrey & Jankowich, 2016*).

**Table 6  Mean differences in lung function by gender, ethnicity, and employment using independent *t*-test.**

| Variables | *n* | Mean (SD) | df | T-statistic | *p*-value |
|---|---|---|---|---|---|
| **Gender** | | | | | |
| **FVC** | | | | | |
| Male | 104 | 3.5 (0.73) | 196 | 12.477 | <0.001 |
| Female | 94 | 2.4 (0.45) | | | |
| **FEV1** | | | | | |
| Male | 104 | 2.9 (0.77) | 196 | 10.398 | <0.001 |
| Female | 94 | 2.0 (0.40) | | | |
| **FEV1/FVC*** | | | | | |
| Male | 104 | 0.8 (0.09) | | 1.542 | <0.123 |
| Female | 94 | 0.8 (0.07) | | | |
| **%FVC** | | | | | |
| Male | 104 | 80.2 (12.29) | 196 | 2.224 | 0.027 |
| Female | 94 | 76.5 (11.06) | | | |
| **%FEV1** | | | | | |
| Male | 104 | 81.6 (13.97) | 196 | 2.074 | 0.039 |
| Female | 94 | 77.7 (11.67) | | | |
| **FEF 25–75%** | | | | | |
| Male | 104 | 3.2 (1.22) | 196 | 5.613 | <0.001 |
| Female | 94 | 2.4 (0.80) | | | |
| **FEF** | | | | | |
| Male | 104 | 7.8 (2.29) | 196 | 10.756 | <0.001 |
| Female | 94 | 4.9 (1.42) | | | |
| **Ethnicity** | | | | | |
| **FVC** | | | | | |
| Malay | 172 | 3.0 (0.83) | 196 | 1.739 | 0.084 |
| Non-Malay | 26 | 2.7 (0.74) | | | |
| **FEV1** | | | | | |
| Malay | 172 | 2.5(0.72) | 196 | 1.704 | 0.090 |
| Non-Malay | 26 | 2.4 (0.69) | | | |
| **FEV1/FVC*** | | | | | |
| Malay | 172 | 0.8 (0.08) | | 0.154 | 0.877 |
| Non-Malay | 26 | 0.8 (0.10) | | | |
| **%FVC** | | | | | |
| Malay | 172 | 79.1 (11.91) | 196 | 2.155 | 0.032 |
| Non-Malay | 26 | 73.8 (10.42) | | | |
| **%FEV1** | | | | | |
| Malay | 172 | 80.7 (13.00) | 196 | 2.801 | 0.006 |
| Non-Malay | 26 | 73.2 (11.48) | | | |
| **FEF 25–75%** | | | | | |
| Malay | 72 | 2.9 (1.12) | 196 | 2.435 | <0.016 |
| Non-Malay | 26 | 2.3 (0.97) | | | |

**Table 6** (*continued*)

| Variables | n | Mean (SD) | df | T-statistic | p-value |
|---|---|---|---|---|---|
| **FEF** | | | | | |
| Malay | 172 | 6.6 (2.42) | 196 | 1.608 | 0.109 |
| Non-Malay | 94 | 5.8 (2.23) | | | |
| **Employment** | | | | | |
| **FVC** | | | | | |
| Employed | 130 | 3.1 (0.84) | 196 | 4.25 | <0.001 |
| Unemployed | 68 | 2.6 (0.66) | | | |
| **FEV1** | | | | | |
| Employed | 130 | 2.6 (0.73) | 196 | 3.997 | <0.001 |
| Unemployed | 68 | 2.2 (0.61) | | | |
| **FEV1/FVC**[*] | | | | | |
| Employed | 130 | 0.8 (0.07) | | 0.637 | 0.524 |
| Unemployed | 68 | 0.8 (0.09) | | | |
| **%FVC** | | | | | |
| Employed | 130 | 79.4 (12.03) | 196 | 1.633 | 0.104 |
| Unemployed | 68 | 76.5 (11.31) | | | |
| **%FEV1** | | | | | |
| Employed | 130 | 80.8 (12.76) | 196 | 1.593 | 0.113 |
| Unemployed | 68 | 77.7 (13.43) | | | |
| **FEF 25–75%** | | | | | |
| Employed | 130 | 2.9 (1.10) | 196 | 2.525 | 0.012 |
| Unemployed | 68 | 2.53 (1.10) | | | |
| **FEF** | | | | | |
| Employed | 130 | 6.9 (2.52) | 196 | 3.974 | <0.001 |
| Unemployed | 68 | 5.6 (1.93 | | | |
| **Smoking** | | | | | |
| **FVC** | | | | | |
| Smoker | 50 | 3.4 (0.72) | 196 | 4.707 | <0.001 |
| Non-smoker | 148 | 2.8 (0.80) | | | |
| **FEV1** | | | | | |
| Smoker | 50 | 2.7 (0.70) | 196 | 3.361 | 0.001 |
| Non-smoker | 148 | 2.4 (0.70) | | | |
| **FEV1/FVC**[*] | | | | | |
| Smoker | 50 | 0.8 (1.03) | | 2.775 | 0.006 |
| Non-smoker | 148 | 0.8 (0.67) | | | |
| **%FVC** | | | | | |
| Smoker | 50 | 80.2 (12.91) | 196 | 1.254 | 0.211 |
| Non-smoker | 148 | 77.8 (11.44) | | | |
| **%FEV1** | | | | | |
| Smoker | 50 | 79.6 (15.45) | 196 | −0.96 | 0.923 |
| Non-smoker | 148 | 79.8 (12.18) | | | |
| **FEF 25–75%** | | | | | |
| Smoker | 50 | 2.9 (1.17) | 196 | 0.936 | 0.350 |
| Non-smoker | 148 | 2.8 (1.10) | | | |

**Table 6** (*continued*)

| Variables | *n* | Mean (SD) | df | T-statistic | *p*-value |
|---|---|---|---|---|---|
| **FEF** | | | | | |
| Smoker | 50 | 7.1 (2.45) | 196 | 2.260 | 0.025 |
| Non-smoker | 148 | 6.2 (2.36) | | | |

**Notes.**
df = degree of freedom.
*FEV1/FVC was analyzed using Mann–Whitney U test due to non-normal distribution, hence standardized test statistic was shown instead of t-statistics.

**Table 7** Correlation analysis between urinary heavy metal and other covariates using Pearson's correlation.

| Variable | N | Log urinary Cd | | | Sqrt urinary Pb | | |
|---|---|---|---|---|---|---|---|
| | | r | p | $r^2$ | r | p | $r^2$ |
| Smoke duration | 50 | .431 | 0.030 | 0.1858 | .263 | 0.780 | 0.0692 |
| Cigarette smoke/day | 50 | .200 | 0.165 | 0.0400 | .281 | 0.048 | 0.0790 |
| Age | 198 | .293 | <0.001 | 0.0858 | .009 | 0.903 | 0.0001 |
| Years living in Klang Valley | 198 | .093 | 0.197 | 0.0086 | -.118 | 0.101 | 0.0139 |
| Air quality (PM10) | 198 | .233 | 0.001 | 0.0543 | .521 | <0.001 | 0.2714 |
| Air quality (PM2.5) | 198 | .179 | 0.012 | 0.0320 | .342 | <0.001 | 0.1170 |
| BMI | 198 | .154 | 0.030 | 0.0237 | -.088 | 0.216 | 0.0077 |
| Log household income | 151 | -.251 | 0.002 | 0.0630 | .002 | 0.979 | <0.0001 |

Based on the reference values in the Michigan Occupational Safety and Health Administration (MIOSHA) references and a recent study (*Ke et al., 2015*), 12.0% of the study participants displayed high urinary Cd (>2 µg/L). Another study in a rural area located in the outskirt of the Klang Valley (*Adnan et al., 2012*) that used the same reference level reported a slightly higher prevalence of 14.7%. This urban-rural discrepancy contrasted with another study that reported a higher level of serum/ urinary Cd among the urban population compared to the rural community (*Alvarez, 2015*). However, the assumption of the urban population would commonly experience a higher Cd exposure than the rural population might not always be accurate as certain rural populations might also be exposed to other sources of Cd as shown in a recent study (*Ashar, Wulandari & Susana, 2018*).

Based on the US CDC, elevated Pb refers to a level of > 50.0 µg/L. In this study, besides none of the participants recorded elevated urinary Pb, majority of participants showed Pb levels lower than 10 µg/L. The average body burden of Pb among the Klang Valley residents in this study was much lower than two decades ago (50.26 µg/L) reported in a previous study (*Hashim et al., 2000*; *Ikeda et al., 2000*). The difference could be attributed to the total abolishment of leaded fuel usage worldwide in the late 1990s that tremendously reduced the environmental lead pollution and exposure among humans. This was supported by a recent study that recorded a marked reduction in atmospheric lead in Canadian cities for the past four decades since the abolishment of leaded gasoline (*Bagur & Widory, 2020*).

Next, log urinary Cd was also negatively correlated with a few lung function parameters in this study, including FVC, FEV1, %FVC, %FEV1, and FEF 25–75%. Nevertheless, the strength of correlation was weak with coefficient values ranging from −.141 to −.227. The

**Table 8** Association between urinary heavy metal and sociodemographic factors using compare mean analysis (independent *t*-test and one-way ANOVA).

| Variables | N (%) | Mean (SD) | df | t-value | p-value |
|---|---|---|---|---|---|
| **Log Urinary Cd** | | | | | |
| **Gender** | | | | | |
| Male | 104 (52.0) | −2.4 (0.62) | 196 | −1.55 | 0.123 |
| Female | 96 (48.0) | −2.3 (0.70) | | | |
| **Ethnicity** | | | | | |
| Malay | 174 (87.0) | −2.4 (0.66) | 196 | −0.348 | 0.728 |
| Non-Malay | 26 (13.0) | −2.3 (0.68) | | | |
| **Employment status** | | | | | |
| Employed | 132 (66.0) | −2.4 (0.69) | 196 | 0.565 | 0.960 |
| Unemployed | 68 (34.0) | −2.4 (0.60) | | | |
| **Highest Education level**[*] | | | | | |
| Primary | 12 (6.0) | −2.1 (0.45) | 2,195 | F = 7.775 | 0.001 |
| Secondary | 69 (34.5) | −2.2 (0.74) | | | |
| Tertiary | 119 (59.5) | −2.5 (0.59) | | | |
| **Hobby/job related to oil painting** | | | | | |
| Yes | 14 (7.0) | −.2.5 (0.59) | 196 | −0.524 | 0.601 |
| No | 186 (93.0) | −2.4 (0.67) | | | |
| **Hobby/job related to gardening** | | | | | |
| Yes | 58 (29.0) | −2.2 (0.67) | 196 | 2.875 | 0.004 |
| No | 142 (71.0) | −2.5 (0.64) | | | |
| **Smoking** | | | | | |
| Yes | 50 (25.0) | −2.3 (0.72) | 196 | 0.481 | 0.631 |
| No | 148 (75.0) | −2.4 (0.63) | | | |
| **(Square root) Urinary Pb** | | | | | |
| **Gender** | | | | | |
| Male | 104 (52.0) | 0.8 (0.35) | 196 | 0.909 | 0.364 |
| Female | 96 (48.0) | 0.8 (0.27) | | | |
| **Ethnicity** | | | | | |
| Malay | 174 (87.0) | 0.8 (0.30) | 196 | −2.300 | 0.023 |
| Non-Malay | 26 (13.0) | 0.9 (0.40) | | | |
| **Employment status** | | | | | |
| Employed | 132 (66.0) | 0.7 (0.32) | 196 | −2592 | 0.010 |
| Unemployed | 68 (34.0) | 0.9 (0.30) | | | |
| **Highest Education level**[*] | | | | | |
| Primary | 12 (6.0) | 0.8 (0.28) | 2,195 | F = 0.664 | 0.516 |
| Secondary | 69 (34.5) | 0.8 (0.31) | | | |
| Tertiary | 119 (59.5) | 0.8 (0.32) | | | |
| **Hobby/job related to oil painting** | | | | | |
| Yes | 14 (7.0) | 0.7 (0.32) | 196 | −1.320 | 0.188 |
| No | 186 (93.0) | 0.8 (0.31) | | | |

**Table 8** (*continued*)

| Variables | N (%) | Mean (SD) | df | t-value | p-value |
|---|---|---|---|---|---|
| **Hobby/job related to gardening** | | | | | |
| Yes | 58 (29.0) | 0.8 (0.26) | 196 | 1.539 | 0.125 |
| No | 142 (71.0) | 0.8 (0.35) | | | |
| **Smoking** | | | | | |
| Yes | 50 (25.0) | 0.8 (0.34) | 196 | −0.005 | 0.996 |
| No | 150 (75.0) | 0.8 (0.31) | | | |

Notes.
*Analysis of "highest education level" variable use one-way ANOVA test and yielded F-statistics.

**Table 9   Predictors of FEV1% and FVC% based on multivariate analysis.**

| | SLR | | | MLR | | | |
|---|---|---|---|---|---|---|---|
| Variables | b | (95% CI) | p-value | Adj. b | (95% CI) | t-stat | p-value |
| **FEV1%** | | | | | | | |
| Log Urinary Cd | −3.76 | (−6.48, −1.03) | 0.007 | −3.21 | (−6.29, −0.14) | −2.07 | 0.040 |
| Ethnicity (Non-Malay) | −7.56 | (−12.88, −2.24) | 0.006 | −8.48 | (−14.46, −2.5) | −2.80 | 0.006 |
| Education (Tertiary) | 6.09 | (2.46, −9.71) | 0.001 | 5.35 | (1.14, 9.56) | 2.51 | 0.013 |
| Education (Secondary) | −5.02 | (−8.80, −1.24) | 0.009 | – | – | – | – |
| Gender (Male) | 3.82 | (0.19, 7.45) | 0.039 | – | – | – | – |
| Household income (RM) | 2.84 | (0.94, 5.58) | 0.043 | – | – | – | – |
| Smoke duration (years) | −0.39 | (−7.34, −0.53) | 0.025 | – | – | – | – |
| **FVC%** | | | | | | | |
| Log Urinary Cd | −4.07 | (−6.52,−1.61) | 0.001 | −3.28 | (−6.14, −0.41) | −2.26 | 0.025 |
| Age | -0.18 | (−0.29, −0.06) | 0.002 | −0.20 | (−0.35, −0.04) | −2.49 | 0.014 |
| Ethnicity (Non-Malay) | -5.32 | (−10.19,−0.45) | 0.032 | −7.40 | (−12.9, −1.81) | 2.62 | 0.010 |
| Gender (Male) | 3.71 | (0.42,7.00) | 0.027 | – | – | – | – |
| Household income (RM) | 2.93 | (0.40,5.46) | 0.023 | – | – | – | – |
| Education (Secondary) | −4.62 | (−8.06,−1.19) | 0.009 | – | – | – | – |
| Education (Tertiary) | 5.35 | (2.05,8.65) | 0.002 | – | – | – | – |

Notes.
SLR, simple linear regression; MLR, multiple linear regression.

negative correlation between urinary Cd and FEV1 in this study aligned with the findings in other studies (*Huang et al., 2016*; *Lampe et al., 2008*). Similarly, the negative correlation between urinary Cd and FVC also mirrored the results of other studies (*Cetintepe et al., 2019*; *Pan et al., 2020*) even although serum Cd was used instead of urinary Cd. In contrast, there was no correlation between urinary Cd and FEV1/ FVC in the study, in contrast with two other studies (*Leem et al., 2015*; *Yang et al., 2019*). A recent study also reported a negative correlation between urinary Cd and FEF 25–25% (*Pan et al., 2020*). Although our study used urinary Cd level as a proxy to measure the Cd body burden, multivariate analysis showed that urinary Cd remained a significant predictor of %FEV1 and %FVC in the final model, thus indicating the important role of Cd in causing lung function impairment. Possible biological mechanism of the effect of Cd exposure on lung function might be explained by the role of Cd acting as a mediator for lung parenchymal damage though disruption of body immune response, specifically triggering modification of mucosal, adaptive, and innate immune responses in lung parenchyma which lead to increase susceptibility to further lung damage and infection risk (*Hossein-Khannazer et al.,*

**Table 10 Predictors of log Urinary Cd and Urinary Pb based on multivariate analysis (Multiple Linear Regression).**

| Variables | SLR | | | MLR | | | |
| --- | --- | --- | --- | --- | --- | --- | --- |
| | b | (95% CI) | *p*-value | Adjb | (95% CI) | t-stat | *p*-value |
| **Urinary Cd** | | | | | | | |
| Age | 0.01 | (0.007, 0.020) | <0.001 | 0.023 | (0.006, 0.04) | 2.825 | 0.007 |
| 3-month mean PM$_{2.5}$ | 0.05 | (0.011, 0.079) | 0.009 | 0.111 | (0.024, 0.19) | 2.592 | 0.013 |
| 3-month mean PM$_{10}$ | 0.03 | (0.013, 0.048) | 0.001 | – | – | – | – |
| Smoking duration | 0.02 | (0.008, 0.037) | 0.003 | – | – | – | – |
| Education (Tertiary) | −0.36 | (−0.540, −0.178) | <0.001 | – | – | – | – |
| Education (Secondary) | 0.30 | (0.109, 0.489) | 0.002 | – | – | – | – |
| household income | −0.23 | (−0.371, −0.93) | 0.001 | – | – | – | – |
| Hobby of gardening | 0.29 | (0.092, 0.491) | 0.004 | – | – | – | – |
| BMI | 0.02 | (0.001, 0.032) | 0.035 | – | – | – | – |
| **Urinary Pb** | | | | – | | | |
| unemployed | 0.12 | (0.290, 0.214) | 0.010 | | – | - | – |
| 3-month mean PM$_{2.5}$ | 0.04 | (0.025, 0.056) | <0.001 | 0.088 | (0.051, 0.124) | 4.853 | <0.001 |
| 3-month mean PM$_{10}$ | 0.03 | (0.025, 0.039) | <0.001 | – | – | – | – |
| Cigarette per day | 0.02 | (0.001, 0.032) | 0.048 | – | – | – | – |
| Height | 0.54 | (0.050, 1.028) | 0.031 | – | – | – | – |

**Notes.**

SLR, simple linear regression; MLR, multiple linear regression.

*2020*). Apart from that, Cd can cause mitochondrial impairment and lipid accumulation in lung cells, indicating long term bioaccumulation and exposure to higher dose may hasten the damage process in the lung tissue (*Hu et al., 2019*).

On the other hand, the square root of urinary Pb was not significantly correlated with most of the lung function parameters, contradicting the results in most of the published studies (*Madrigal et al., 2018*; *Pan et al., 2020*; *Xiao et al., 2016*; *Yang et al., 2019*). However, two other studies echoed our findings in which no correlation was detected between Pb and impaired lung function (*Nti, 2015*; *Xu et al., 2021*). Different target populations can explain the variation between the study findings. Notably, a significant correlation between serum Pb with worsening lung function was commonly detected among coal workers with occupational exposure but not the general population (*Gogoi et al., 2019*).

Lastly, education, ethnicity, and age were significant predictors of lung function. Tertiary education level led to a better FVC, likely due to better health awareness and preventive behavior among this subgroup of participants. Another study also revealed that lower education and household income levels were associated with low awareness level and preventive behavior towards air pollution (*Low et al., 2020*). Ethnicity and age remained significant predictors even after adjustment. However, as the reference range of the current population is dynamic and ever-changing, a periodical revision from time to time is necessary (*Jian et al., 2017*). Lastly, the 3-month average PM2.5 concentration was significantly correlated with both urinary Cd and Pb. This concurs with the study in China (*Wu et al., 2021*) in which PM2.5 level was associated with urinary Cd level among children.

The limitations of this study include the lack of other environmental samples to prove the epidemiological linkage between exposure and lung function. Secondly, this study

was prone to operator-dependent error as the results were highly dependent on the participants' understanding and their efforts to perform satisfactory lung function tests. For instance, a participant might have deliberately or unintentionally performed poorly on the lung function test, thus producing "false positive impairment" data that led to inaccurate results and conclusions. Furthermore, this study was not spared of recall bias, especially in collecting data on participants' lifestyle and socio-economic characteristics. Another limitation was, the urinary Cd used in this analysis was not corrected with the urinary creatinine level. Furthermore, authors acknowledge that measuring urinary Pb in a non-contaminated setting was also one limitation that might contribute to the non-significance correlation between urinary Pb concentrations and lung function parameter in this study. The author also acknowledges this final model limitation, which did not include other pollutants such as nitric oxide or other heavy metals. In order to yield a better final model, more factors or pollutant should be added in multivariable analysis as confounders. Lastly, this study's sample size was rather small compared with other large-scale surveys. Nevertheless, this study contributed to the literature on the importance of early lung screening.

## CONCLUSION

Urinary Cd displayed a negative linear relationship with lung function, particularly %FVC and %FEV1. In contrast, urinary Pb did not display any significant correlations with lung function. Furthermore, education level, ethnicity and age were significant predictors of lung function. This study might strengthen the existing link between heavy metals exposure and lung function impairment.

## ACKNOWLEDGEMENTS

Special thanks to the research volunteers involved in the data collection process and the hosts of various community events who facilitated the data collection.

### Funding

This research was funded by Universiti Kebangsaan Malaysia research grant. The funders had no role in study design, data collection and analysis, decision to publish, or preparation of the manuscript.

### Grant Disclosures

The following grant information was disclosed by the authors:
Universiti Kebangsaan Malaysia research grant.

### Competing Interests

The authors declare there are no competing interests.

## Author Contributions

- Ammar Amsyar Abdul Haddi conceived and designed the experiments, performed the experiments, analyzed the data, prepared figures and/or tables, authored or reviewed drafts of the article, and approved the final draft.
- Mohd Hasni Ja'afar conceived and designed the experiments, authored or reviewed drafts of the article, and approved the final draft.
- Halim Ismail conceived and designed the experiments, authored or reviewed drafts of the article, and approved the final draft.

## Human Ethics

The following information was supplied relating to ethical approvals (i.e., approving body and any reference numbers):

This study received ethical approval from the University Kebangsaan Malaysia (UKM) Research Ethical Committee (Ethical Approval reference: UKM PPI/111/8/JEP-2018699 FF-2019-043, 10 January 2019).

## Data Availability

The raw data is available as a Supplemental File.

## Supplemental Information

Supplemental information for this article can be found online at http://dx.doi.org/10.7717/peerj.13845#supplemental-information.

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
