# Peer review of "Association between lung function impairment with urinary heavy metals in a community in Klang Valley, Malaysia"

_PeerJ, doi:10.7717/peerj.13845_

## Round 0.1 · original submission · Major Revisions

Dear Authors, please make the suggested revisions by reviewers or write a detailed rebuttal on a point-by-point basis.

Reviewer 1 ·

Basic reporting

In general, this is an interesting research. The authors aim to explore the prevalence of lung function impairment as well as its association with urinary heavy metal levels and other influencing factors among the community in Klang Valley, Malaysia. However, there are points should be improved.

The article included sufficient introduction and background to demonstrate how the work fits into the broader field of knowledge. Relevant prior literature appropriately referenced.

Experimental design

1. The statistical used in this is not clear. Due to the aim of the study was to explore association between the lung function and urinary heavy metal levels. Thus, the multivariable regression analysis must be used. The others pollutant such as PM10, PM 2.5, O3, CO, NO etc. must be included as confounding factors in the analysis.
2. The results were not correlated with the objective of the study, For example, Table 5, 6. Why you present the mean differences in lung function by education level and mean differences in lung function by gender, ethnicity, and employment. The rational for presenting these results must be provided. I recommend including these parameters (education level, gender, ethnicity, and employment) as confounding factors in the multivariable regression analysis.
3.The inclusion and exclusion criteria is not clear. Why the subjects with history of chronic respiratory diseases including asthma and COPD not excluded from the study. Please clarify.

Validity of the findings

The results were not correlated with the objective of the study, For example, Table 5, 6. Why you present the mean differences in lung function by education level and mean differences in lung function by gender, ethnicity, and employment. The rational for presenting these results must be provided. I recommend including these parameters (education level, gender, ethnicity, and employment) as confounding factors in the multivariable regression analysis.

Additional comments

Abstract
- mean age of 38.4, the SD of age should be provided.
- “full vital capacity” should be changed to “forced vital capacity”
- “force expiratory volume at 1 second” should be changed to “force expiratory volume in the first second”
- “force expiratory flow between 25-75% FVC” should be changed to “force expiratory flow between 25-75% of FVC”
Introduction
- The full term of Cd and Pb should be provided.
Methods
- line 96, sample size calculation, please provide the parameter that account for sample size calculation
- line 112, “CHESTRAPH” should be changed to “CHESTGRAPH”
- line 117, please provide the reference of ATS guidelines.
Statistical Analysis
- You mentioned that “Pearson's correlation test were used to analyze the effect of continuous variables on lung function”. I think the Pearson's correlation showed only the correlation but not effect.
Definition
- “Forced Expiratory Volume in one second” should be changed to “Forced Expiratory Volume in the first second”.
Results
- line 186, please check the unit of PEF, L/hour is correct?
- in the table 2,3, and 4. The decimal should be consistency.
Discussion
- In overall, the discussion section was not consisted with the main findings of this study. Thus, I recommend editing the discussion section.

Annotated reviews are not available for download in order to protect the identity of reviewers who chose to remain anonymous.

Reviewer 2 ·

Basic reporting

• Regarding phrases of “Urinary Cd (or Pb),” you used “Urinary” and “urinary” inconsistently in the manuscript. Please use the same expression. I would like to recommend that authors use “urinary.”
• In the explanation of Table 10 (e.g., lines 256 and 259), multivariate analysis models were described as the “final model,” but it seems unclear. Please make those explanations clearer with more specific expressions.

Experimental design

• The authors used urinary concentrations of cadmium and lead as biomarkers to assess exposure levels. However, urinary lead concentration may not be suitable for the assessment, especially in non-contaminated settings. This is a very important limitation in this study. Also, although urinary lead concentration did not show any correlations with lung function parameters, the null results might be because of the limitation (Lines 312–319).
• Lines 78–80: if the authors wanted to study other sources of heavy metal exposure, all analyses should be performed separately by smoking status (smoker/non-smoker).
• Did you correct urinary heavy metal concentrations for urine dilution effects? In usual, urine specific gravity or urinary creatinine concentration is used for the correction.
• You apply log-transformation for urinary Cd concentration, while you use square route-transformation for urinary lead concentration. Please add explanation of the reasons why you used two different methods of transformation separately.
• Line 250: Could you describe which variables you considered as confounders in the multivariate models.
• Line 252: Could you more deeply explain the results of interactions?

Validity of the findings

• Line 48–50 in the Abstract: the explanation seems unclear. Urinary cadmium concentration must be a biomarker of cadmium exposure level.
• Lines 299–311: The authors only discuss the comparison of results of associations of cadmium exposure and lung function parameters between the present and previous studies. Please add the explanation of the possible biological mechanisms of the effects of Cd exposure on lung function.

Additional comments

This paper discusses the associations between cadmium and lead exposure and lung function impairment. The paper provides very interesting data but it still needs a considerable revision to be acceptable for the PeerJ. Please see my comments and suggestions below, which would contribute to improving your paper.

Reviewer 3 ·

Basic reporting

no comment

Experimental design

The author has to explicitly mention how the study fills the gap in the introduction section

Validity of the findings

The author need to revise the conclusion of the article

Annotated reviews are not available for download in order to protect the identity of reviewers who chose to remain anonymous.

---

## Round 0.2 · Minor Revisions

Please revise as suggested by reviewers.

Reviewer 1 ·

Basic reporting

no comment

Experimental design

no comment

Validity of the findings

no comment

Additional comments

no comment

Reviewer 2 ·

Basic reporting

I think authors have improved the manuscript well based on the comments from reviewers.
Before acceptance, please confirm the manuscript overall to avoid any typos (e.g., "mitrchondrialm"=>"mitochondrial").

Experimental design

No additional comments.

Validity of the findings

no additional comments.

Additional comments

no additional comments.

Reviewer 3 ·

Basic reporting

no comment

Experimental design

no comment

Validity of the findings

no comment

Additional comments

See attached PDF

Annotated reviews are not available for download in order to protect the identity of reviewers who chose to remain anonymous.

---

## Round 0.3 · accepted · Accept

Dear Authors,

Your manuscript is acceptable for publication in its current form.